# Perceived family support status and associated factors among people with hypertension in Nekemte City public hospitals, Western Ethiopia

**Zewude Mulatu Geleta** [ID]1*, **Dereje Chala Diriba**2, **Desalegn Emana Jabana**2

**1** Wollega University Comprehensive Specialized Hospital, Institute of Health Sciences, Wollega University, Nekemte, Ethiopia, **2** Department of Nursing, Institute of Health Sciences, Wollega University, Nekemte, Ethiopia

\* zewudemulatu2019@gmail.com

## Abstract

### Background

Though previous studies reported the advantages of family in treating people with hypertension, evidence indicates that family support among people with hypertension was poor.

To the best of the researchers' knowledge, no study examined the status of perceived family support and associated factors among people with hypertension in Ethiopia. Therefore, this study assessed perceived family support status and associated factors among people with hypertension in Nekemte City public hospitals, Western Ethiopia,

### Methods

A hospital based cross sectional study on 422 selected adults with hypertension was conducted in public hospitals in Nekemte from June to August 2023. A systemic random sampling technique was used to select eligible participants. Bivariate and multivariable logistic regression analyses were carried out, and variables with a p-value < 0.25 in the bivariate analysis were included in the multivariable analysis to identify factors associated with perceived family support at a P-value < 0.05.

### Results

Two hundred ninety two (70.4%), 95% CI (65.7%, 74.7%), of people with hypertension reported perceived strong family support. Having a family history of hypertension [AOR = 3.6, 95% CI (1.266, 10.34)], involvement of families during patient education on HTN [AOR = 3.305, 95% CI (2.306, 8.363)], and good interaction with family [AOR = 2.45, 95% CI (1.002, 5.32)] were positively associated with perceived family support. Whereas, patients who are poorly knowledgeable about hypertension [AOR = 0.3, 95% CI (0.001, 0.542)] were negatively associated with perceived family support,

**Data availability statement:** All relevant data are within the paper and its Supporting Information files.

**Funding:** The author(s) received no specific funding for this work.

**Competing interests:** The authors have declared that no competing interests exist.

**Abbreviations:** AOR, Adjusted Odds Ratio; BP, Blood Pressure; CI, Confidence Interval; COR, Crudes odds Ratio; DALYS, Disability Adjusted Life Years; HTN, Hypertension; MmHg, Millimeters of Mercury; MRN, Medical Record Number; NCD, Non-Communicable Disease; NCSH, Nekemte Comprehensive Specialized Hospital; OPD, Outpatient Department; PSS-fa, Perceived Social Support Family Scale; SPSS, Statistical package for the Social sciences; WHO, World Health Organization; WUCSH, Wollega University Comprehensive Specialized Hospital

## Conclusion

Two hundred ninety two people with hypertension reported perceived strong family support. Having a family history of hypertension, involvement of families during patient education on hypertension, and patients' interaction with family were positively statistically associated with perceived family support and patients who are poorly knowledgeable about hypertension were negatively associated with perceived family support. Hospitals, along with other stakeholders, should strengthen family support by involving families in health education in line with hypertension patients' lifestyles.

## Introduction

Hypertension (HTN) is a silent killer disease and it is one of the most significant risk factors for cardiovascular morbidity and mortality resulting from target organ damage to the heart, brain, kidney, and/or eyes [1]. The World Health Organization (WHO) reports a staggering rise in hypertension cases, affecting one in three adults globally [2]. Uncontrolled hypertension leads to 13.5% of premature deaths and 6% of lost years of life due to death or disability [3].

Self-care practices are essential in hypertension management. One essential strategy for promoting practice management is family support. HTN management cannot only be undertaken by patients and health care providers alone but also requires support from family members [4]. Globally, family support is essential as the primary source of emotional, material, informational, and financial support to hypertension patients [4].

Perceived family support is defined as the extent to which an individual perceives that a support system is meeting his or her care needs in a support network provided by a family network consisting of father/mother, daughter/son, spouse, and other relatives or a group of people related by birth, marriage, adoption, or other relationship who give support to the hypertensive patient [5].

Family members were the best choice for a supervisor, and they could ensure accessibility for a large proportion of patients. Family members who reside with or are close to the patient have emotional ties to them and are legally permitted to make choices on their behalf [6]. Family support plays a critical role in helping persons with hypertension maintain and control their blood pressure and get it back to normal [7]. Families are seen by patients as a source of comfort, understanding, and support, and their involvement is essential for patients and healthcare services in the fight against medication and food non-adherence [8].

Family members can help hypertensive patients in several ways, such as by providing them with positive encouragement, exchanging information, offering support during times of need, and providing general health care, which includes diet and exercise advice as well as reminders to take medications and schedule checkups [9].

There has been evidence to suggest a correlation between family support and positive health outcomes [10]. According to certain research, those who experienced greater degrees of family support stuck to their treatment regimen and had lower systolic blood pressure [5,8,11,12].

Some studies noted the beneficial influence of family support on embracing healthy behaviors, including quitting smoking, losing weight, adhering to a balanced diet, taking regular blood pressure readings, going to the doctor regularly, exercising, and using drugs [3,10,13].

Despite WHO recommendation of family support for hypertensive patients, [14], there is suboptimal family support. Limited studies were conducted to assess the perceived family support for hypertensive patients. There were inconclusive perceived family support statuses among hypertensive patients. Previous studies conducted in Turkey, India and Nigeria range between 22.95% and 79.4% [15–18]

Several factors affect the support status of the family members among people with HTN, including economic status of the family [19], families' knowledge about the disease [20], cohesive or conflict status of family [11], patients interaction with family, families' age, families education status, length of the patient stayed with families, families job, families gender [21], family member with HTN, family member who had suffered complications from HTN, patient age, patient education status [22]. There is no study showing perceived family support status and associated factors among people with HTN in Ethiopia. Hence, this study aimed to examine the status of perceived family support and associated factors among people with HTN.

## Methods and materials

### Study design, area, and period

A prospective cross-sectional study design was conducted from June to August 2023 at Wallaga University Comprehensive Specialized Hospital (WUCSH) and Nekemte Comprehensive Specialized Hospital (NCSH). Both hospitals are located in Nekemte city, 331 km west of the capital city of Ethiopia. The hospitals provide health care for hypertension and other diseases. They share the same catchment population, which is 3.5 million people and serve as referral for 10 million people living in Western Ethiopia. [23]. Five hundred and four hundred ten adults with hypertension were attending medical follow-up at NCSH and WUCSH during the study period respectively.

### Participants

Data collectors approached people with hypertension in the hospital when they wait for monthly medical follow-ups. Subjects were included if they 1) had been diagnosed with hypertension, 2) were aged 18 years and above, 3) were attending an outpatient department, 4) were in a stable medical condition, and 5) were cognitively intact (as determined in their medical records). They were excluded if they were 1) refused to participate in the study, and 2) were from prison custody because they would not have adequate time to provide their responses.

### Sample size estimation

The final required sample size was estimated using a single population proportion formula because the estimated sample sizes using variables found predictors from previous studies are less than the estimated sample size using a single population proportion. Since there was no previous study that reported the prevalence of perceived family support among people with hypertension in Ethiopia, we assumed 50% prevalence, 95% confidence level and 50% margin of error. Considering a 10% non-response, a sample size of 422 people with hypertension was required.

### Sampling technique

A systematic random sampling technique was applied to select the study participants in both hospitals. Every second from the sampling frame in hospitals was approached to participate in the study.

### Variables

**Dependent variable.** Perceived family support status

**Independent variables.** Socio-demographic variables include age, gender, marital status, education level, occupation status, residence, income, and family size. Family-related variables like gender, marital status, availability of a family member who is living with HTN, income, education level and occupation status, and accompanying the patient on education about hypertension, and the patient-related variables, such as disclose of HTN, duration since diagnosis, availability of comorbidity, interaction with family, and knowledge of hypertension were collected.

## Data collection tool

The questionnaire was developed including socio-demographic variables, items on family-related characteristics, and patient-related characteristics, and perceived social support. Items to assess the socio-demographic characteristics, family and most of the patient-related characteristics were developed by the authors referring to other literature.

**Patient's knowledge about hypertension.** It was assessed using an assessment of knowledge about hypertension among hypertensive patients. The tool comprises 18 items with different response rates. While all items are positively scored, items 2, 4–9 and 12 were reversely scored [24]. The mean was calculated. The people's knowledge about hypertension was categorized into good knowledge if the mean score is nine or above and poor if the mean score is less than nine.

**Patient family interaction.** It is assessed using the family communication scale. The tool has ten items and is scored with a 5 point Likert scale (1 strongly disagree to 5 strongly agree) [25]. The patient family interaction was calculated in mean. If the mean score of interaction is seven and above, it indicates a good interaction whereas a score of less than seven indicates poor. The Cronbach alpha of the scale was 0.97, indicating excellent internal consistency.

**Perceived family support.** The participant's perceived family support was assessed using the Perceived Social Support Family Scale (PSS Fa) [26]. It is a 20-item self-reported scale which assesses the subject's perceived support from the family. The item has three possible responses "Yes," "No" or "I don't know." The responses of each participant summed together and a score of 11 or greater is interpreted as strong perceived family support and a score less than 10 is interpreted as weak perceived family support [27]. The scale has acceptable reliability and validity. The internal consistency of the original scale showed excellent reliability [26]. The translation into the target language and pretested on 21 participants was done.

## Data collection procedure

The data were collected through face-to-face interviews. One day training was given for data collectors to enable them to understand each item in the scales and the techniques of effective interviews. Subjects were contacted when they were waiting to see a doctor for their medical check-up in the hospital. After explaining the benefits and risks of the study, data collectors assessed the people with hypertension for their eligibility. To reduce information bias, family member who accompanied with patients were asked to leave the area.

## Statistical analysis

The statistical analyses were performed using SPSS version 26. Descriptive analyses were computed for socio-demographic, family and patient-related characteristics, and the level of perceived family support. The frequency and percentage were calculated for categorical variables, whereas the pooled scores were obtained for the level of knowledge, the interaction and perceived family support. The bivariate analyses were performed to identify candidate variables for a multivariable analysis.

A p value < 0.25 on the bivariate analyses were candidates for the multivariable analysis not to miss important variables, to control for the possible effects of confounding, and determine associated factors on perceived family support among people with HTN. A backward logistic regression model was used to identify factors associated with the perceived family support.

Model assumptions like multicollinearity and model fitness were checked by a variation inflation factor (VIF) which ranges from the lowest 1.105 to the highest 6.326, and Hosmer and Lemshew which is 0.7, respectively. A crude and adjusted odds ratio together with the corresponding 95% confidence intervals was computed. A p-value <0.05 and confidence intervals excluded zero were used to decide statistical significance. The value of AOR and its 95% confidence intervals were reported in the results.

## Ethical considerations

Ethical clearance was obtained from the Research Ethics Review Committee of Wollega University (Reference Number: WU/RD/673/2023). Permission to collect data at the hospitals was obtained from hospital administration. Informed written consent was obtained from each study participant. Each completed questionnaire was coded, and the code was used for data entry to ensure anonymity of the participants.

## Results

### Socio-demographic characteristics of the participants

A total of 415 participants participated in the study, showing a 98% response rate. The mean age was 56.5 (*SD* ± 12.6) years. More than half (57.8%) were males, more than three quarters (80.5%) were married, and slightly more than one-third (34.2%) of the participants had attended higher education. Slightly more than one-third (35.7%) had five and above family size (Table 1).

### Family-related characteristics

The median family monthly income was 4000 with an interquartile range of 2500–6500 ETB and 129 (31%) of the study reported their monthly family income was between 3201 and 5250 ETB. Over half (62.4%) of them had no family history of HTN, and more than half (55.7%) of them were females (Table 2).

**Patient-related characteristics.** More than two-thirds (67.2%) of participants lived with HTN for more than 2 years, and nearly half (44.6%) had other chronic comorbidities, of which diabetes was the most common (61.1%). Over two-thirds (68.7%) had good patient-family interaction, and 69.2% of them had good knowledge about hypertension (Table 3).

**Perceived family support status among people with HTN.** More than two-thirds (70.4%), [95% CI (65.7%, 74.7%)] of the study participants had perceived strong family support [Fig 1].

### Factors associated with perceived family support among people with HTN

The bivariate analyses showed that age, gender of the family primary caregiver, education level of family primary caregiver, place of residence, patient monthly income, family history of HTN, the income of family primary caregiver, occupation of family primary caregiver, marital status of family primary caregiver, involvement of families during patient education on HTN, duration of HTN, presence of chronic comorbidity, interaction of patient with family, and knowledge of patient about HTN were associated with the perceived family support (Table 4).

Variables with a p-value less than 0.25 [28] were included in the multivariable logistic regressions. Four variables: family history of HTN, involvement of families during patient education on HTN, patient interaction with family, and knowledge of the patient about HTN, were s significantly associated with perceived family support among people with HTN (Table 4).

The multivariable logistic regression showed that the odds of strong perceived family support among patients and their families involved during patient education on HTN were 3.3 times [AOR=3.3, (95%CI (2.306–8.363))] more likely as compared to families not involved during patient education on HTN.

The odds of perceived family support among families who have a history of HTN were 3.6 times [AOR=3.6, (95%CI (1.266–10.34))] more likely than those who have no family history of HTN. The odds of getting perceived family support

**Table1. Socio-demographic characteristics of participants (n =415).**

| Variables | Category | Frequency | Percent (%) |
|---|---|---|---|
| Age (in years) | 30–34 | 13 | 3.1 |
| | 35–39 | 33 | 8.0 |
| | 40–44 | 24 | 5.8 |
| | 45–49 | 51 | 12.3 |
| | 50–54 | 51 | 12.3 |
| | 55–59 | 65 | 15.6 |
| | 60–64 | 54 | 13.0 |
| | 65–69 | 44 | 10.6 |
| | >=70 | 80 | 19.3 |
| Gender | Male | 240 | 57.8 |
| | Female | 175 | 42.2 |
| Marital status | Married | 334 | 80.5 |
| | Single | 20 | 4.8 |
| | Divorced | 12 | 2.9 |
| | Widowed | 49 | 11.8 |
| Education level | No formal education | 98 | 23.6 |
| | Primary | 66 | 15.9 |
| | Secondary | 109 | 26.3 |
| | Higher | 142 | 34.2 |
| Occupation status | Employed | 114 | 27.5 |
| | Laborer | 17 | 4.1 |
| | Merchants | 68 | 16.4 |
| | Farmer | 86 | 20.7 |
| | Other* | 130 | 31.3 |
| Place of residence | Urban | 276 | 66.5 |
| | Rural | 139 | 33.5 |
| Monthly income (in ETB) | 601–1650 | 39 | 9.4 |
| | 1651–3200 | 113 | 27.2 |
| | 3201–5250 | 115 | 27.7 |
| | =>5251 | 148 | 35.7 |
| Family size | Two | 77 | 18.5 |
| | Three | 69 | 16.6 |
| | Four | 121 | 29.2 |
| | Five and above | 148 | 35.7 |

*Homemaker, Student, and Retired; ETB: Ethiopia Birr; 1 Ethiopian Birr =0.008 USD*

among patients who have poor knowledge of HTN were 0.3 times [AOR=0.3, (95%CI (0.001–0.542))] less likely as compared to patients who have good knowledge of HTN. The odds of perceived family support among patients who have good interaction with their family were 2.45 times [AOR = 2.45, (95%CI (1.002–5.32))] more likely as compared to patients who have poor interaction with their family (Table 4).

## Discussion

The study was conducted to assess perceived family support and associated factors among people with HTN. Accordingly, the finding showed that strong perceived family support was reported among 70.4% of participants. Family history of

**Table 2. Family-related characteristics of participants (n = 415).**

| Variables | Category | Frequency | Percent (%) |
|---|---|---|---|
| Family history of HTN | Yes | 156 | 37.6 |
| | No | 259 | 62.4 |
| **Income of family primary caregiver (ETB) | 601–1650 | 43 | 10.4 |
| | 1651–3200 | 116 | 28.0 |
| | 3201–5250 | 129 | 31.0 |
| | =>5251 | 127 | 30.6 |
| Education of family primary caregiver | No formal education | 69 | 16.6 |
| | Primary | 62 | 15 |
| | Secondary | 113 | 27.2 |
| | Higher | 171 | 41.2 |
| Occupation of family primary caregiver | Employed | 151 | 36.4 |
| | Labourer | 11 | 2.6 |
| | Merchants | 126 | 30.4 |
| | Farmer | 51 | 12.3 |
| | Other* | 76 | 18.3 |
| Gender of family primary caregiver | Male | 184 | 44.3 |
| | Female | 231 | 55.7 |
| Marital status of family's primary caregiver | Married | 328 | 79.0 |
| | Single | 64 | 15.5 |
| | Divorced/separated | 13 | 3.1 |
| | Widowed | 10 | 2.4 |
| Families involved in patient education on HTN | Yes | 223 | 53.7 |
| | No | 192 | 46.3 |

*Homemaker, Student, and Retired; ** 1 Ethiopian Birr = 0.008 USD*

**Table 3. Patient-related characteristics of the participants (n = 415).**

| Variables | Category | Frequency | Percent (%) |
|---|---|---|---|
| Disclose your HTN to the family | Yes | 407 | 98.1 |
| | No | 8 | 1.9 |
| Duration of diagnosis | less than or equal to 2 years | 136 | 32.8 |
| | greater than 2 years | 279 | 67.2 |
| Chronic comorbidity other than HTN | Yes | 185 | 44.6 |
| | No | 230 | 55.4 |
| If yes what are other comorbidities | Diabetic mellitus | 113 | 61.1 |
| | Heart failure | 55 | 29.7 |
| | Renal failure | 10 | 5.4 |
| | Other* | 7 | 3.8 |
| Patient-family interaction | Good interaction | 285 | 68.7 |
| | Poor interaction | 130 | 31.3 |
| Knowledge about hypertension | Good knowledge | 287 | 69.2 |
| | Poor knowledge | 128 | 30.8 |

*Bronchial Asthma, Cancer, and Retroviral infection*

**Perceived Family support status among people with HTN**

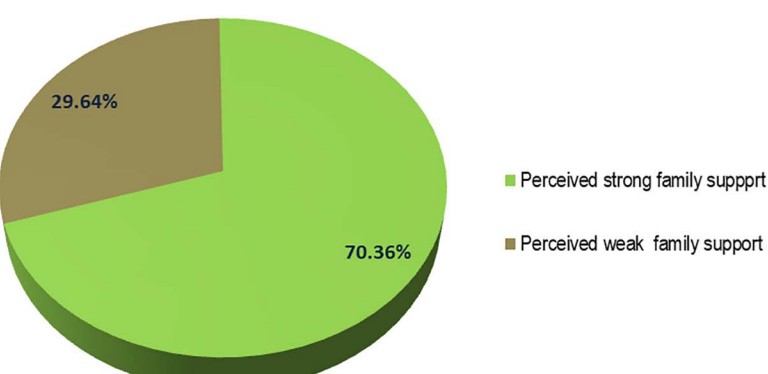

**Fig 1. Perceived family support status among people with HTN in Nekemte city public hospitals, Western Ethiopia, 2023.**

HTN, families involved during patient education on HTN, patient interaction with family, and knowledge of people with HTN about HTN were significantly associated with perceived family support among people with HTN.

The proportion of study participants who perceived strong HTN family support (i.e., 70.4%) in the current study is almost consistent with the study finding from Nigeria (69.7%) [29], this shows Africans have naturally rich family support networks. The reasons

for this may be explained by the extended family system in our culture. It is known that in family-centered societies, people tend to gain major support from family as noted in the previous research works [30, 31] and both studies used the same family support assessment scale (PSS-Fa). But greater than the rates reported in Malaysia 63.3% [32]. This difference may be related to Ethiopians' culture, which emphasizes familial ties and the responsibility to look out for one another, as seen in the coffee ceremony, their holiday celebrations, and their mourning.

On the other hand, the proportion is less than the finding of the study from Kenya 82% [27], Iran 82.9% [33], and Indian 91% [34]. Variations may be due to differences in sociocultural aspects of HTN patients of Iran, India, Kenya, and Ethiopia. The effectiveness of some family support may depend on factors outside the family environment, and community resources for HTN management and that may be related to the cultural and family integration of the study population.

The result of this study shows that those who reported a family history of HTN, involvement of families during patient education on HTN and good interaction with family are more likely to get perceived family support. This means people with HTN whose family member had HTN, good interaction with family and involvement of families during patient education are getting more advice, helpful information, caring, and trust from the family member which might help to cope with HTN [35,36]. Whereas, people with HTN who are poor knowledgeable about HTN are less likely to have perceived family support when compared to the others in the category.

The finding of this study indicated that family history of HTN was associated with perceived family support among people with HTN. The study finding confirmed that the odds of having perceived family support were higher among participants with family history of HTN as compared with those reported no family history of HTN. This finding is supported with the finding the study conducted in Nigeria [37], which indicated that, having a family history of HTN had significant relation with perceived family support from family. This might show that the longer caring for and living with people with hypertension, the more knowledge gained from experience during daily care.

Human behaviour is formed based on the results of experience which were first modified to increase effectiveness and the longer the human behavior becomes habitual and individuals increasingly try to modify behavior that is suitable for

**Table 4. Bivariate and Multivariable analyses of variables with perceived family support.**

| Variables | Category | Perceived family support status | | COR (CI 95%) | AOR (CI 95%) |
|---|---|---|---|---|---|
| | | **Strong** | **Weak** | | |
| Age category | 30–34 | 6(46.2) | 7(53.8%) | 0.2(0.06,0.73) | 0.34(0.02,4.05) |
| | 35–39 | 14(42.4%) | 19(57.6%) | 0.18(0.08,0.45) | 0.159(0.013,2.005) |
| | 40–44 | 14(58.3%) | 10(41.7%) | 0.35(0.13,0.93) | 0.14(0.013,1.5) |
| | 45–49 | 37(72.5%) | 14(27.5%) | 0.66(0.29,0.7) | 0.38(0.05,3.09) |
| | 50–54 | 35(68.6%) | 16(31.4%) | 0.55(0.24,0.73) | 0.77(0.12,4.9) |
| | 55–59 | 48(73.8%) | 17(26.2%) | 0.7(0.3,0.84) | 0.47(0.08,2.9) |
| | 60–64 | 37(68.5%) | 17(31.5%) | 0.5(0.25,0.6) | 0.54(0.08,3.64) |
| | 65–69 | 37(84.1%) | 7(15.9%) | 1.3(1.09,3.5) | 2.1(0.24,5.6) |
| | ≥70 | 64(80%) | 16(20%) | 1 | 1 |
| Place of residence | Urban | 57(20.7%) | 219(79.3%) | 1 | 1 |
| | Rural | 66(47.5%) | 73(52.5%) | 0.29(0.19,0.45) | 1.6 (0.4. 6.15) |
| Patient monthly income | 601–1650 | 23(59%) | 16(41%) | 0.32(0.15,0.67) | 0.3 (.05, 1.8) |
| | 1651–3200 | 75(66.4%) | 38(33.6%) | 0.44(0.25,0.78) | 0.38(.09, 1.7) |
| | 3201–5250 | 73(63.5%) | 42(36.5%) | 0.39(0.22,0.68) | 0.47(0.1, 1.9) |
| | =>5251 | 121(81.8%) | 27(18.2%) | 1 | 1 |
| Family history of HTN | Yes | 140(89.7%) | 16(10.3%) | 6.16(3.5,10.9) | 3.6(1.3,10.34)** |
| | No | 152(58.7%) | 107(41.3%) | 1 | 1 |
| Family primary care giver income category | 601–1650 | 27(62.8%) | 16(37.2%) | 0.26(0.12,0.58) | .79(0.1, 5.9) |
| | 1651–3200 | 65(56%) | 51(44%) | 0.1(0.105,0.37) | 1 (.22, 5.23) |
| | 3201–5250 | 90(69.8%) | 39(30.2%) | 0.357(0.189,0.672) | 1.78(0.37,8.5) |
| | =>5251 | 110(86.6%) | 17(13.4%) | 1 | 1 |
| Family education status | No formal education | 36(52.2%) | 33(47.8%) | 0.29(0.16,0.53) | 0.46(0.05,4.06) |
| | Primary | 44(71%) | 18(29%) | 0.65(0.34,0.86) | 2.29(0.53,5.09) |
| | Secondary | 77(68.1%) | 36(31.9%) | 0.57(0.33,0.98) | 1.53(0.29,8) |
| | Higher education | 135(79%) | 36(21%) | 1 | 1 |
| Occupation of family Primary caregiver | Employed | 27(17.9%) | 124(82.1%) | 1 | 1 |
| | Laborer | 5(45.5%) | 6(54.5%) | 0.26(0.07,0.92) | 1.1(0.09, 9) |
| | Merchants | 44(35%) | 82(65%) | 0.4(0.2,0.7) | 3(0.13, 4.96) |
| | Farmer | 22(43%) | 29(57%) | 0.29(0.14,0.57) | 2(0.08, 6.7) |
| | Other | 25(33%) | 51(67%) | 0.4(0.24,0.8) | 4(0.09, 3.9) |
| Primary care giver gender of family | Male | 99(58%) | 72(42%) | 0.36(0.26,0.89) | 2(0.7, 6.9) |
| | Female | 193(79%) | 51(21%) | 1 | 1 |
| Marital status of family | Married | 77(23.5%) | 251(76.5%) | 1 | 1 |
| | Single | 35(54.7%) | 29(45.3%) | 0.254(0.146, 0.443) | 0.369(0.095,1.426) |
| | Divorced | 6(46.2%) | 7(53.8%) | 0.36(0.12, 0.8) | 0.8(0.01,2.63) |
| | Widowed | 5(50%) | 5(50%) | 0.3(0.09, 0.4) | 0.08(0.004,1.507) |
| Families involved in patient education on HTN | Yes | 180(80.7%) | 43(19.3%) | 3(1.9,4.6) | 3.3(2.3,8)** |
| | No | 112(58.3%) | 80(41.7%) | 1 | 1 |
| Duration of diagnosis | ≤ 2 years | 71(52.2%) | 65(47.8%) | 0.3(0.18,0.45) | 0.8(0.26, 2.7) |
| | > 2 years | 221(79.2%) | 58(20.8%) | 1 | 1 |
| Presence of chronic comorbidities | Yes | 116(62.7%) | 69(37.3%) | 0.5(0.34,0.8) | 0.9(0.3, 2.7) |
| | No | 176(76.5%) | 54(23.5%) | 1 | 1 |
| Patient-family interaction | Good | 200(70.2%) | 85(29.8%) | 5(3,8) | 2(1.002,5.32)** |
| | Poor | 40(30.8%) | 90(69.2%) | 1 | 1 |

*(Continued)*

**Table 4.** (Continued)

| Variables | Category | Perceived family support status | | COR (CI 95%) | AOR (CI 95%) |
|---|---|---|---|---|---|
| | | **Strong** | **Weak** | | |
| Knowledge about hypertension | Good | 12(4.2%) | 275(95.8%) | 1 | 1 |
| | Poor | 111(87%) | 17(13%) | 0.007(0.003,0.014) | 0.3(0.001, 0.542)** |

*1-Referance category; ** Significant at p-value less than 0.05*

various situations [38]. Thus, for those who have a family history of HTN their family supporters may have more experience, knowledge, and confidence in providing support for simpler tasks like reminding date of appointment for follow-up, minimizing salt in wat, and hypertensive self-management information.

The current study also showed that higher odds of perceived family support for study participants with involvement of families during patient education on HTN. The finding is supported by a study finding from Iran [39]. The results revealed that involvement of families during patient education plays a significant role in HTN management, by encouraging the patients to adhere to medication, low sodium diet and medical appointments [39]. Because family member education with the presence of patients may promote adopting a healthier lifestyle and better managing BP in patients with HTN. It will eventually reduce the medical cost for patients and healthcare systems. There is a significant reduction of salt intake among patients receiving health education with family member [40].

In this study, there was a significant association between the interaction of people with HTN with their families and perceived family support. Accordingly, those who had good interaction of people with HTN with their family were found to be having more perceived family support compared to those having poor interaction with people with HTN with their family. This study is supported by study conducted in Indonesia [21].

The more the interaction of people with HTN with their family, the more likely it is to improve family decision-making in determining the control and treatment of people with hypertension. When there is interaction there is communication, closeness, and support.

A strong family support relationship can increase emotions and enthusiasm which leads to some positive health outcomes [41].

According to this study, Patient knowledge about HTN was found to be significantly associated with perceived family support. Patients with poor knowledge about HTN were less likely to get family support as compared to patients with good knowledge about HTN. This study is in line with a study conducted in Indonesia [42]. Poor knowledge about HTN and its treatment creates a poor understanding and confusion about the treatment and the disease condition. Knowledge has been an important domain in establishing the overt behavior of an individual [7]. People with HTN who have poor knowledge about HTN may not try avoiding or minimizing anything that might cause/exacerbate a disease in the family.

### Strengths and limitations of the study

This study was the first to be carried out on perceived family support among people with HTN in Ethiopia. This study solely relied on reports from the patients themselves. Thus, the findings may be prone to recall bias. Eliciting only family acceptable responses and hence, may overestimate the perceived family support status. The study design is cross-sectional and it does not establish cause and effect relationship between outcome variables and predictors.

### Conclusions

Most adults living with hypertension perceived strong family support for their diseases. Their perception of family support was predicted by having a family caregiver, family history of HTN, involvement of families during patient education, and

patients' interaction with family. However, the patient's poor knowledge of hypertension has negatively affected the perception of the family support. Further study is warranted to establish the association between these variables and people living with hypertension's perception of family support.

## Supporting information

**S1 File. English Version Questionnaire.**
(DOCX)

## Acknowledgments

We thank the study participants for their participation in the study.

## Author contributions

**Conceptualization:** Zewude Mulatu Geleta.

**Data curation:** Zewude Mulatu Geleta, Dereje Chala Diriba, Desalegn Emana Jabana.

**Formal analysis:** Zewude Mulatu Geleta, Dereje Chala Diriba.

**Methodology:** Desalegn Emana Jabana.

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
