## [Decision Letter · Decision Letter 0]

15 Sep 2024

PONE-D-24-13805Perceived family support status and associated factors among people with hypertension in Nekemte town public hospitals, Western Ethiopia, 2023PLOS ONE

Dear Dr. Geleta,

Thank you for submitting your manuscript to PLOS ONE. After careful consideration, we feel that it has merit but does not fully meet PLOS ONE’s publication criteria as it currently stands. Therefore, we invite you to submit a revised version of the manuscript that addresses the points raised during the review process.

The manuscript can benefit from the reviewer's comment and suggestion, I advise you to correct the manuscript based on the comment provided below. 

We look forward to receiving your revised manuscript.

Kind regards,

Dawit Gebeyehu Getachew, MPH

Academic Editor

PLOS ONE

Journal Requirements:

Reviewers' comments:

Reviewer's Responses to Questions

**Comments to the Author**

1. Is the manuscript technically sound, and do the data support the conclusions?

Reviewer #1: No

Reviewer #2: No

2. Has the statistical analysis been performed appropriately and rigorously? 

Reviewer #1: No

Reviewer #2: Yes

3. Have the authors made all data underlying the findings in their manuscript fully available?

Reviewer #1: Yes

Reviewer #2: No

4. Is the manuscript presented in an intelligible fashion and written in standard English?

Reviewer #1: No

Reviewer #2: No

5. Review Comments to the Author

Reviewer #1: Review report on paper titled “Perceived family support status and associated factors among people with hypertension in Nekemte town public hospitals, Western Ethiopia, 2023”

Comments

Title

#Need to be more concise, potentially excluding “2023, status”.

Who are these people? the support may vary based on actors like gender or age

Abstract

The background section requires revision to provide a more comprehensive background as it looks introduction.

The methodology needs to be summarized, and there are concerns about saturating a sample size of 422 with random selection.

The prevalence rate may be 69.2%?.

Justification is needed for the frequency, duration, and specific areas of education or may counseling

The author should strive for a straightforward, non-technical conclusion and recommendation, avoiding statistical language that may not be easily understood by policymakers.

Introduction

Excessive length and lack of concision

Paragraphs that are not well-linked or cohesive

Missing information on the prevalence and associated factors of the subject

May be luck of a conceptual framework to guide the reader and analysis

Absence of a clearly defined objective for the study

Materials and methods

Is the cross-section design prospective or retrospective?

……excluded patients who are unable to hear and/or communicate. What action you would have been taken to include this group as they have the right to include their opinion? May the guardian approach be able to address this concern?

Nice Sampling technique and procedure

Data collection tool, …….Then the questionnaire was pre-tested at Ambo University Referral Hospital on 5% of the sample size which was 21 patients ; make it brief and it was an exciting to read travelling a very long distance to conduct a pre-test.

Limit redundancy from data collection produce and Data quality assurance

Nicely leveled Data processing and analysis

Lengthy write-up under Ethical considerations

The variation inflation factor (VIF) value (1-6) included the range (5 ≤ VIF < 10) that suggests the existence of moderate multicolinerity. What measures the authors have taken to overcome this?

Report

Table 4 with larger decimal places

Discussion

Improve the coherence

Strengths and limitations of the study

Need standard write-up

Reviewer #2: The document is full of grammatical illusions, and can lead readers to initially interpret them in a way that seems correct, only to reveal a grammatical error upon closer examination. The document may be benefited from major grammatical and punctuation revision. Authors have to be advised on manuscript preparation for publication, word selection and necessary detail needed in manuscript. The manuscript has no line numbers, making it difficult to make reference to sections of the document while reviewing. PDF version was also difficult to make line numbers from the reviewer's copy.

Introduction

Page 3

Paragraph 1: The phrases "silent killer disease" and "time bomb" , in the sentence can be simplified into “Silent killer". The paragraph is drafted with poor flow and transition. It could benefit from better transitions between ideas. For example, linking the rise in hypertension cases to the consequences mentioned would create a smoother flow. Plus the whole paragraph is full of fragmented phrases and needs to be paraphrased.

Paragraph 3: The sentence is quite long and could be broken down for better understanding. Plus, the phrase "support system is meeting his or her care needs in a support network" is repetitive. The term "support network" already implies a system of support

Paragraph 4: “Family members were the best choi ce for a supervisor…” what does it mean? Please take and merge the last sentence with the second or third sentence of the paragraph.

Page 4,

Paragraph 1: “There has been evidence to suggest a correlation between…” make it another paragraph and add with the subsequent paragraph.

Paragraph 4 &5: These two paragraphs complement messages of paragraph 1 on page 3. What is the importance of repeating similar ideas here?

Page 5

Paragraph 1: What does it mean by “poor family functioning”? Is it similar to poor family support/weak family support? Please use similar expression throughout your document

Paragraph 3: “However, several factors affect the support status…” and you have listed several factors affecting family supports. In light of this, the issue is not a lack of evidence regarding its importance. Instead, the challenge lies in translating this evidence into practice, highlighting the need for implementation research; understanding why effective family support is not consistently applied in practice is crucial. Why you prefer to repeat it?

Study period: No need for exact details here. Say June to August 2023 vs June 1st to August 30 th, 2023.

Page 6: population

Try to avoid avoidable naming of hospitals repeatedly.

Inclusion and exclusion criteria

“The participants were included in this study if they were 18 years old and above and excluded patients who are unable to hear and/or communicate.” Rewrite this statement carefully

Sample size determination

Why did you use a single population proportion formula for calculating sample size while you have conducted analytical study?

Please know how far detail sample size calculation and sampling technique should be in manuscript prepared for publication.

Page 8

Data collection tool: Please put references from which you adopted and adapted your tools

More than half page is composed of a single paragraph. Separate into paragraphs.

“Initially, knowledge assessment scale questions were t wenty, but after the pretest two questions were removed because they were unclear for pa tients. During pretest I gave fifteen minutes to complete the questionnaire, but it took twenty” What is the importance of reporting these? Please try to adhare manuscript preparation guideline for scientific community

Please merge and describe your study variables, operational definition and measurement well.

Page 10: Data processing and analysis

The whole page is composed of a single paragraph. Please separate into paragraphs. Please take ideas above, “The data was coded, entered into Epidata version 4.6, and transferred to SPSS version 26 for analysis”, to measurement and operational definition section. It would also be good if you advised on how composite score of items measured on likert scale are important for converting such items into category. Plus be advised on how the analysis should be detailed. Same statement will also need grammatical revision.

Page 13/Result

Socio-demographic factors

“A total of 422 hypertensive subjects were recruited from the two hospitals from June 1 st to August 30th . Among the participants invited to participate, the majority of them 415(9 8%) gave their complete responses, and 7 (2%) of them were non response rate because t hey were not volunteered”. Remove unnecessary detail like study period and summarize only response rate.

“…and the confidence Int erval of the age of the study participants was (55.37, 57.73)”, please be advised how CI should have to be reported. Plus, describe all variables in tables.

Over half (57.8%) were male: words and %; 185(44. 6%) had other chronic comorbidities: number and %; diabetes mellitus (61.1%) was the most co mmonly reported comorbidity: only %. Please make your writing follow similar reporting format throughout the document.

gender of family primary ca regiver, education status of family primary caregiver, income of family primary caregiver, occupation of family primary caregiver, marital status of family primary caregiver…..these variables are vague for the readers to understand and seem non reputable. Make them reputable.

Factors associated with perceived family support among people with HTN

“getting perceived family support”, is a vague phrase and difficult to understand. Please make it in line with your measurement and operational definition.

Discussion

Rewrite the first paragraph and separate into statements.

Paragraph 2: It is good that that you discussed possible justification for the difference of your finding from previous literatures. However, most of your claims lack references and make your justification questionable. Please support your claim with references throughout your discussion. For example; “The reasons for this can be explained by the extended family system in our environment”. May you add reference for this statement that indicates extended family of Nigeria and Ethiopia? Add references for others claim in your discussion too.

Page 23, first paragraph: make it two paragraphs.

Strengths and limitations of the study: Merge Strengths and limitations and write in the statement form

Conclusion: Please don’t depend on other persons’ findings to conclude your finding. Just conclude your work as per your objectives. No need to have separate recommendation in manuscript rather than preparing your conclusion in a way it informs recommendation.

6. PLOS authors have the option to publish the peer review history of their article (what does this mean? ). If published, this will include your full peer review and any attached files.

**Do you want your identity to be public for this peer review?** For information about this choice, including consent withdrawal, please see our Privacy Policy .

Reviewer #1: No

Reviewer #2: **Yes: ** Gachana Midaksa

---

## [Author Response · Author response to Decision Letter 1]

30 Nov 2024

I like the comments of both reviewers

---

## [Decision Letter · Decision Letter 1]

27 Dec 2024

PONE-D-24-13805R1Perceived family support status and associated factors among people with hypertension in Nekemte city public hospitals, Western Ethiopia, 2023PLOS ONE

Dear Dr. Geleta,

Thank you for submitting your manuscript to PLOS ONE. After careful consideration, we feel that it has merit but does not fully meet PLOS ONE’s publication criteria as it currently stands. Therefore, we invite you to submit a revised version of the manuscript that addresses the points raised during the review process.

We look forward to receiving your revised manuscript.

Kind regards,

Dawit Gebeyehu Getachew, MPH

Academic Editor

PLOS ONE

**Journal Requirements:**

**Additional Editor Comments:**

Dear author, We have received the reviewer evaluation report regarding your response to the review report on the first round review and the revised manuscript. Although you have made improvements on the manuscript, the second reviewer requested that you missed addressing the comment on the revised manuscript; you only provided the response on the author response to the reviewer. Therefore, please try to incorporate all the required amendments in the revised manuscript.

Reviewers' comments:

Reviewer's Responses to Questions

**Comments to the Author**

1. If the authors have adequately addressed your comments raised in a previous round of review and you feel that this manuscript is now acceptable for publication, you may indicate that here to bypass the “Comments to the Author” section, enter your conflict of interest statement in the “Confidential to Editor” section, and submit your "Accept" recommendation.

Reviewer #1: All comments have been addressed

Reviewer #2: (No Response)

2. Is the manuscript technically sound, and do the data support the conclusions?

Reviewer #1: Yes

Reviewer #2: Partly

3. Has the statistical analysis been performed appropriately and rigorously? 

Reviewer #1: Yes

Reviewer #2: Yes

4. Have the authors made all data underlying the findings in their manuscript fully available?

Reviewer #1: No

Reviewer #2: No

5. Is the manuscript presented in an intelligible fashion and written in standard English?

Reviewer #1: Yes

Reviewer #2: No

6. Review Comments to the Author

**Reviewer #1:**  Dear authors, your paper now looks fine for dissemination at global scale. Thank you for the revision, and again consider these minor point.

# Include reference for this sentence in paragraph one of introduction “Effective management is important to reduce morbidity and mortality”.

#If you did not encounter individuals who could not hear during data collection, it would not be appropriate to list them as an exclusion criterion unless you had a specific rationale for excluding them that was relevant to your study design.

**Reviewer #2: ** Authors partly modified the manuscript. However, the authors responses are too shallow and did not cover my previous comments. Some comments like research gaps, sample size calculation and measurement need justification in manuscript rather than trying to respond for the reviewers. The manuscript may be benefited from revision considering the previous review results.

7. PLOS authors have the option to publish the peer review history of their article (what does this mean? ). If published, this will include your full peer review and any attached files.

**Do you want your identity to be public for this peer review?** For information about this choice, including consent withdrawal, please see our Privacy Policy .

Reviewer #1: No

Reviewer #2: No

---

## [Author Response · Author response to Decision Letter 2]

28 Feb 2025

I like the comments of both reviewers

---

## [Editor Report · Decision Letter 2]

3 Mar 2025

Perceived family support status and associated factors among people with hypertension in Nekemte city public hospitals, Western Ethiopia

PONE-D-24-13805R2

Dear Dr. Geleta,

We’re pleased to inform you that your manuscript has been judged scientifically suitable for publication and will be formally accepted for publication once it meets all outstanding technical requirements.

Kind regards,

Dawit Getachew Gebeyehu, MPH

Academic Editor

PLOS ONE
---

## [Editor Report · Acceptance letter]

PONE-D-24-13805R2

PLOS ONE

Dear Dr. Geleta,

I'm pleased to inform you that your manuscript has been deemed suitable for publication in PLOS ONE. Congratulations! Your manuscript is now being handed over to our production team.

Kind regards,

on behalf of

Mr. Dawit Getachew Gebeyehu

Academic Editor

PLOS ONE